# Determinants of Response at 2 Months of Treatment in a Cohort of Pakistani Patients with Pulmonary Tuberculosis

**DOI:** 10.3390/antibiotics11101307

**Published:** 2022-09-26

**Authors:** Saeed Shah, Asghar Khan, Muhammad Shahzad, Jawahir A. Mokhtar, Steve Harakeh, Zeeshan Kibria, Aneela Mehr, Bushra Bano, Asif Ali, Yasar Mehmood Yousafzai

**Affiliations:** 1Institute of Pathology and Diagnostic Medicine, Khyber Medical University, Peshawar 25120, Pakistan; drsaeedshah86@gmail.com (S.S.); dr.asghar007khan@gmail.com (A.K.); draliasif7@gmail.com (A.A.); 2Institute of Basic Medical Sciences, Khyber Medical University, Peshawar 25120, Pakistan; shahzad.ibms@kmu.edu.pk (M.S.); aneelamehr2287@gmail.com (A.M.); bushybano1994@gmail.com (B.B.); 3School of Biological Sciences, Health and Life Sciences Building, University of Reading, Whiteknights, Reading RG6 6AX, UK; 4Department of Medical Microbiology and Parasitology, Faculty of Medicine, King Abdulaziz University, Jeddah 21589, Saudi Arabia; jmokhtar@kau.edu.sa; 5Vaccine and Immunotherapy Unit, King Fahd Medical Research Center, King Abdulaziz University, Jeddah 21589, Saudi Arabia; 6King Fahd Medical Research Center and Yousef Abdullatif Jameel Chair of Prophetic Medicine Application, Faculty of Medicine, King Abdulaziz University, Jeddah 21589, Saudi Arabia; sharakeh@gmail.com; 7Institute of Public Health and Social Sciences, Khyber Medical University, Peshawar 25120, Pakistan; zeeshankibria@kmu.edu.pk; 8College of Medical Veterinary and Life Sciences, University of Glasgow, Glasgow G12 8QQ, UK; 9Institute of Infection, Immunity and Inflammation, University of Glasgow, Glasgow G12 8QQ, UK

**Keywords:** tuberculosis, prognostic stratification, TB

## Abstract

Mycobacterium tuberculosis infection continues to be a major global challenge. All patients with pulmonary tuberculosis are treated with a standard 6-month treatment regimen. Historical data suggest that even with shortened treatment, most patients achieve long-term remission. Risk stratification is a goal for reducing potentially toxic prolonged treatment. This study aimed to determine the factors associated with the early clearance of sputum acid-fast bacilli (AFB). A total of 297 freshly diagnosed patients with pulmonary tuberculosis were included and enrolled in this study. Information related to their ethno-demographic and anthropometric characteristics was collected. We also assessed their complete blood counts, and blood iron, folate, and vitamin B12 levels. We found that the presence of higher levels of acid-fast bacilli (AFB) in diagnostic sputum microscopy was the single most significant prognostic factor associated with early clearance of sputum AFB after 2 months of treatment. All of our patients achieved treatment success after 6 months of treatment and were disease free. Our results support the data obtained from previous studies indicating that AFB clearance at 2 months is unlikely to be a clinically useful biomarker or indicator for therapeutic stratification. Furthermore, demographic, anthropometric, and nutritional factors are not clinically useful biomarkers.

## 1. Introduction

Tuberculosis (TB), caused by *Mycobacterium tuberculosis*, continues to be one of the leading infectious diseases despite the global efforts that are exerted for its prevention and treatment [1,2]. Recent estimates suggested that around one-fourth of the world population is infected with latent TB with no clinical signs and symptoms, of which 5–10% may develop active TB at some stage of their life [3]. The latest global report on TB (2021) suggested that in 2020, around 9.9 million people were infected with TB, which resulted in the death of around 1.5 million people [4]. The current recommended treatment consists of a 2-month intensive treatment regimen with isoniazid, rifampicin, pyrazinamide, and ethambutol, followed by a 4-month continuation phase with an isoniazid–rifampicin combination for new patients with pulmonary and some cases of extrapulmonary TB [4,5]. Longer treatment is required for some extrapulmonary TB cases (central nervous system, bones, and joints), or multi-drug-resistant TB [6,7].

The current standard 6-month treatment for drug-susceptible tuberculosis is long—especially compared with that for most bacterial infections [8,9]. This exceptionally long treatment poses two main challenges: Firstly, drug compliance in extended home treatment regimens is not followed well by many patients. Despite the focus on directly observed treatment (DOT) programs, studies showed poor drug adherence in many cases. Secondly, there is significant drug toxicity associated with the use of multiple drugs for longer periods. Up to 13% of patients show signs of hepatotoxicity [10], although some studies showed that up to half of patients do not complete the prescribed regimen due to toxicities [11]. Historical data from trials conducted in the 1970s revealed that 2–3 months of treatment resulted in the cure of 70–80% of patients [12], implying that the majority of patients may not need such a prolonged regimen.

There is a need for better risk stratification at the time of diagnosis for a more personalized treatment regimen based on the patient’s individual risk. We asked whether ethno-demographic, anthropometric, or laboratory parameters can predict early clearance of AFB from sputum smears in patients with pulmonary TB treated with a standard regimen. The response was measured at 2 months using sputum microscopy for AFB as a marker for evaluation. Patients were categorized into early responders (no AFB at 2 months) and late responders (residual AFB at 2 months). A six-month follow-up was performed to assess the treatment endpoint. Patient demographic characteristics, as well as anthropometric and hematological parameters, were compared between early and later responders.

## 2. Results

This longitudinal observational study included a total of 297 newly diagnosed, treatment-naïve pulmonary tuberculosis patients. The median age of the participants was 26 years (Q1–Q3 20–50 years) and the men:women ratio was 1.2:1. The median height of the participants was 158.5 cm (Q1–Q3 152.4–167.2 cm), and the median weight was 50 kg (Q1–Q3 41–57). The majority of participants received no formal education (mode 0 years), with the mean years of participant education being 2.6 years (SD ± 4.3). Most participants were Pakistani in origin (96.6%), ethnic Pashtuns (94.9%), and nonsmokers (89.6%). Most participants lived in their own accommodations (77.8%) and had separate toilets (95.6%) and kitchens (82.8%) in their homes. Natural gas was used in most participants’ homes for cooking (62.3%), followed by firewood (28.6%). Basic demographic, anthropometric, and laboratory variables are presented in Table 1.

### 2.1. Diagnostic Sputum AFB Levels Are Significantly Different between Early and Late Responders

After 2 months, follow-up sputum samples were obtained, and microscopy was performed. A total of 288 patients were assessed, out of whom 223 (77.4%) had no detectable AFB in sputum ZN-stain microscopy—these were labeled as early responders. Sixty-five (22.5%) patients had residual AFB and were hence labeled as late responders. After 6 months, all patients had completed treatment and were free of AFB on sputum microscopy. Upon comparison of the mean values and frequencies, early responders had a slightly higher red cell mean corpuscular red volume (MCV) (77.2 ± 11.6 vs. 74.5 ± 9.5 *p*-values 0.028) and lower white cell counts (9.9 ± 3.7 vs. 10.9 ± 4.3 *p*-values 0.047). No significant differences were found in demographic, anthropometric, or biochemical parameters. Remarkably, highly significant differences between early and late responders were observed in the sputum AFB levels at the diagnostic smear. Almost all (78/79 (98%)) patients with a scant AFB load at diagnosis had a negative sputum AFB at 2 months. Similarly, 120/126 (95.2%) patients with an AFB load of 1+ and 25/45 (55.6%) with an AFB load of 2+ had a negative AFB load at 2 months. Only 7/47 (14.7%) patients with an initial sputum AFB of 3+ had a cleared sputum AFB load at the 2-month follow-up (*p*-value < 0.001, chi-square test) (Table 2).

### 2.2. Diagnostic AFB Levels Are Independent Predictors of Early Response

Multivariate regression analysis was performed to assess the determinants of early response to therapy. Early and late responses were taken as the dependent variables. Demographic variables (such as age, sex, income, and type of cooking fuel) and clinical variables (such as comorbidities, and hematological and biochemical parameters) were taken as the independent variables after collinearity diagnostics using tolerance statistics. Collinearity tolerance values were set at >1 and <10. Outliers were manually assessed using scatterplots. The results of multivariate analyses on factors that were associated with treatment response for early clearance of AFB in sputum are listed in Table 3. For the clearance of sputum AFB at 2 months of therapy, s lower AFB burden at diagnosis was a strong independent predictor of early response (odds ratios (2+ AFB, OR 0.01; 95% CI 0.001–0.01), (3+ AFB, OR 0.001 95% CI 0.0002–0.02); *p*-value < 0.001). No other variable in the multivariate model was found to be associated with early clearance of AFB (Table 3).

## 3. Discussion

We set out to investigate the determinants of an early response to TB treatment. In this study, we stratified patients into early and late responders based on the clearance of AFB from sputum smears. We demonstrated that the AFB burden (measured with simple microscopy) predicted early clearance of AFB at two months of treatment. However, both groups successfully completed their treatment and were disease-free at the end of 6 months. In this study, we included several demographic parameters known to be associated with TB prevalence or treatment adherence. These included age, sex, ethnicity, and social factors such as household structure and income. None of these variables had any association with AFB clearance at two months. This indicated that the TB control program and DOTS regimen are effective. Patients from lower socioeconomic strata showed no evidence of noncompliance as documented by AFB clearance. Similarly, hematological parameters, such as the presence of anemia, or circulating white blood cells had no association in the multivariate analysis. Nutritional status is generally considered to be a poor prognostic factor in TB treatment outcomes [13,14]. However, in this study, the levels of body iron, vitamin B12, or folate had no association with treatment outcomes. C-reactive protein, a marker of inflammation, had no association with AFB clearance either. Because all patients were cured, treatment failure could not be determined.

In the past, several approaches have been tried to identify a suitable predictive biomarker of treatment response [15]. In a large meta-analysis, sputum microscopy/culture at 2 months of therapy was found to have low predictive accuracy for treatment failure. For treatment failure, 2-month microscopy had a sensitivity of 57% and specificity of 81% [12]. The authors of a meta-analysis concluded that a 2-month smear and culture did not have enough predictive value of treatment failure. However, this does not detract from the need to find suitable biomarkers for the prediction of disease burden, treatment failure, or relapse.

The reasons for the lack of association between long-term treatment response and initial AFB burden (or early clearance) may be numerous [16]. Recent evidence showed that TB bacteria are sequestered in tissue compartments with poor drug penetration sites, as evidenced in granulomas, which are naturally walled-off compartments, and frequently in fibrosis resulting from inflammation [17]. Novel drug delivery techniques may be an area of focus rather than increasing the number of drugs or prolonging the duration of treatment. Furthermore, TB bacteria showed a biphasic decline in bacterial numbers following ATT. This implies that the majority of bacteria susceptible to treatment are killed, leaving behind more persistent bacteria. These persistent bacteria are in a slow-cycling phase [4]. More studies are required to better elucidate the underlying mechanisms by which these bacteria behave at both the molecular and cellular levels. Furthermore, the drug metabolism of each patient may be different.

So far, TB treatment regimens have been advised as a ‘one size fits all’ approach, overlooking drug pharmacokinetics and pharmacodynamics. The role of drug transporters and metabolizers should be studied. Finally, the number of AFB in sputum at diagnosis or during treatment may not accurately represent the total disease burden. Host factors such as the presence of lymphatic vasculature in different individuals may allow AFB to be excreted in sputum. The role of adhesion molecules that allow bacteria to adhere to tissues or come out in sputum may also play a role. Sputum AFB burden may not reflect the complex disease mechanism involved in tuberculosis. Factors such as drug susceptibility, bacterial pathogenicity, and host immune system are involved [18,19,20]. Biomarkers of subclinical inflammation, such as interferon-gamma should be investigated as markers of treatment response.

The understanding of bacterial infections and host–pathogen interactions has undergone rapid evolution in the past few decades [21,22,23]. In tuberculosis, the use of molecular techniques for diagnosis has made early diagnosis and treatment possible. However, treatment failure and relapse of TB still pose major challenges [24,25]. Consequently, pharmacological agents and regimens have stayed more or less the same. The standard 6-month treatment for drug-susceptible TB is longer for a proportion of patients. Overtreatment in some cases to ensure appropriate treatment for others is the current dogma of treatment [26,27,28,29]. Due to treatment side effects and compliance problems, shortening the treatment regimen is a desirable goal. There is a need for a personalized approach to TB treatment.

## 4. Methodology

### 4.1. Study Design, Subjects, and Inclusion Criteria

The study was approved by the Ethical Review Committee of Khyber Medical University and permitted by the Provincial TB Control Program, Khyber Pakhtunkhwa. This multicenter, longitudinal observational study was carried out from February to October 2018 at the District Tuberculosis Office (DTO) Laboratory Peshawar, Al-Khidmat Hospital Peshawar and District Headquarter Hospital Kohat. Sample size was calculated using OpenEpi (version 3.01) using the formula n = [DEFF × Np(1 − p)]/[(d2/Z21 − α/2 × (N − 1) + p × (1 − p)]. We estimated the frequency of slow response to be 20%, and with a confidence limit of 5%, design effect of 1, confidence level of 95%, and drop-out rate of 20%, we calculated a sample size of 297. Adult (aged 18 years and above) patients who were freshly diagnosed with pulmonary tuberculosis were included and enrolled in this study. The participant information sheet was provided to all the participants to read and was also verbally explained in their native language. Written informed consent was obtained from the participants. Patients with a previous history of tuberculosis or antituberculosis therapy; diagnosed with multidrug-resistant or extrapulmonary tuberculosis; and those with associated comorbidities such as COPD, asthma, ILD, and lung cancer were excluded from the study. Tuberculosis diagnosis was based on sputum smear positivity of at least two sputum samples, collected on the same day for acid-fast bacilli (AFB) using Zeil-Neelson staining. Based on the number of AFB present, the patients were grouped as either scant (1–9 AFB/100 field), + (10–99 AFB/100 field), ++ (1–9 AFB/field), or +++ (10–99 AFB/field) as per the American Thoracic Society guidelines.

### 4.2. Collection of Socio-Demographic and Anthropometric Data and Blood Samples

A structured questionnaire was used to collect information about socio-demographic characteristics such as age, sex, ethnicity, education, occupation, household income, and smoking status from all the participants. Anthropometric measurements were recorded by trained data collectors and followed standard guidelines. To measure height, the participants were asked to stand still with their backs to a wall-mounted stadiometer (Seca, Birmingham, U.K.) and head in the Frankfurt plan position. Height was measured to the nearest 0.1 cm by lowering the measuring device until it gently rested on the scalp without putting pressure. To measure weight, the participants were instructed to remove their shoes, added clothes and any extra weight, and stand on a precalibrated electronic scale (Seca, U.K.). Weight was recorded in kilograms to the nearest 0.1 kg. For consistency, all the measurements were taken three times and the average was recorded. Body mass index (BMI) was calculated as the fraction of weight to the squared height (kg/m^2^).

At least 5 mL of blood was collected from each participant by a trained phlebotomist using standard aseptic techniques. Following collection, blood samples were processed by centrifugation at 4000 rpm for 10 min, and the resulting serum was stored at −80 °C until analysis.

### 4.3. Laboratory Analysis

Laboratory investigations included the complete blood count (CBC) and a range of hematological parameters. The CBC was immediately performed after blood collection on the Sysmex hematology analyzer (XS-500i, Sysmex Europe GmbH, Norderstedt, Germany) following manufacturer instructions and including different parameters such as hemoglobin (g/dL), red blood cells count, mean corpuscular volume, mean corpuscular hemoglobin, hematocrit, white blood cells, and platelets count. Serum iron, ferritin, transferrin, unsaturated iron-binding capacity (UIBC), and C-reactive proteins were analyzed by the electrochemiluminescence (need to confirm) method using a Cobas^®^ e601 (Roche Diagnostics GmbH, Mannheim, Germany) analyzer. The total iron-binding capacity (TIBC) was calculated by summing up the UIBC and serum iron concentration.

### 4.4. Follow Up

All the patients received free, first-line standard drug therapy per guidelines from the national TB control program [30]. The treatment was free of cost and included isoniazid, rifampicin, pyrazinamide, and ethambutol for the first two months and isoniazid and rifampin for the following 4 months. The DOT treatment protocol was followed for all patients. Per the DOT protocol, a responsible person in the neighborhood (local school teacher, madrassah teacher, or an educated neighbor) was designated as an observer. The observer was required to watch the patient ingest every dose. The first follow-up was performed 2 months following the initiation of therapy, and the second at 6 months, upon the completion of therapy. At follow-ups, all the patients had undergone sputum smear microscopy for AFB. Patients were labeled as slow responders when the sputum AFB was still positive for two months. The treatment outcome was declared either cured, treatment completed, treatment failed, deceased, or lost to follow-up as per NTCP guidelines.

### 4.5. Statistical Analysis

Data were entered using Microsoft Excel 2016 (Microsoft, Los Angeles, CA, USA) and exported to Statistical Package for Social Sciences (SPSS) version 25 (IBM, Armonk, NY, USA). g for continuous variables and frequencies for discrete variables. Multivariate regression was used to calculate OR for response at 2 months of ATT, adjusting for the following potential confounding factors: age, sex, BMI, monthly household income, income spent on food, smoking history, AFB burden at diagnosis, Hb levels, WBC counts, serum ferritin, iron, TIBC and transferrin levels, vitamin B12 and folate levels, and serum CRP levels.

## 5. Conclusions

Early clearance of sputum AFB in pulmonary TB appears to be directly correlated with at-presentation sputum AFB levels and is unlikely to be useful for therapeutic stratification. The level of sputum AFB at diagnosis is unlikely to be a clinically useful predictor of treatment response. Other demographic, anthropometric, and laboratory variables, although perhaps individually important, are not independently associated with early clearance.

## Figures and Tables

**Table 1 antibiotics-11-01307-t001:** Basic demographic and anthropometric characteristics of the participants (*n* = 297).

Variable		Median	Q1, Q3
Age (years)		26.0	(20, 50)
Height (cm)		158.5	(152, 167.2)
Weight (kg)		50.0	(41, 57)
BMI		19.39	(16.8, 22.4)
Formal education (years)		0	(0, 5)
Monthly household income (PKR)		25,000	(10,000, 20,000)
Monthly income (PKR) spent on food		15,000	(10,000, 20,000)
Variable		No.	% Age
Sex	Male	132	(44.4%)
	Female	165	(55.6%)
History of smoking	Yes	20	(89.6%)
	No	258	(6.9%)
	Ex-smoker	10	(3.5%)
Comorbidities	Yes	44	(15.4%)
	No	241	(84.5%)
Country of birth	Pakistan	287	(96.6%)
	Afghanistan	10	(3.4%)
First language	Chitrali	1	(0.3%)
	Hindko	14	(4.7%)
	Pashtu	282	(94.9%)
Religion	Hinduism	1	(0.3%)
	Islam	296	(99.7%)
Homeownership status	Madrassah	1	(0.3%)
	Own home	231	(77.8%)
	Rented	65	(21.9%)
Separate bathroom	Yes	284	(4.4%)
	No	13	(95.6%)
Separate kitchen	Yes	246	(82.8%)
	No	51	(17.2%)
Type of fuel for cooking	Electricity	2	(0.7%)
	Firewood	83	(28.6%)
	Natural gas	185	(63.8%)
	Combined	29	(6.9%)
AFB burden at diagnosis	Scanty	79	(26.6%)
	+1	126	(42.4%)
	+2	45	(15.2%)
	+3	47	(15.8%)
Characteristic		Mean	Std. Deviation
Hemoglobin (g/dL)		12.1	(2.40)
RBC counts (10⁶/µL)		4.58	(0.96)
Mean corpuscular volume (fL)		76.35	(10.92)
Mean corpuscular hemoglobin (pg)		26.05	(4.8)
Mean corpuscular hemoglobin concentration (g/dL)		34.2	(2.35)
Platelet counts (10³/µL)		380.5	(214.75)
White cell counts (10³/µL)		9.70	(4.20)
Hematocrit (%)		35.5	(6.56)
Serum ferritin (ng/mL)		293.45	(384.38)
CRP levels		29.05	(50.13)
Serum vitamin B12 levels		383.20	(455.75)
Serum folate levels		6.6	(9.48)
Total iron0binding capacity		240	(92.76)
Serum transferrin		131	(52.75)
Serum iron		47.47	(28.99)

Q1 = 25th percentile, Q3 = 75th percentile.

**Table 2 antibiotics-11-01307-t002:** Comparison of demographic, anthropometric, and laboratory findings of early and late responders (*n* = 288). Patients were grouped into early and late responders based on presence or absence of AFB in 2-month sputum smears, and their demographic, anthropometric, and laboratory findings were compared using an independent samples t-test (continuous variables) or chi-square test (categorical variables).

Characteristic		Early Responders (*n* = 223) (Mean ± SD)	Late Responders (*n* = 65) (Mean ± SD)	*p*-Value
Age (years)		34.2 ± 17.9	37.3 ± 19.6	0.234
Height (cm)		159.3 ± 12.3	157.4 ± 11.9	0.282
Weight (kg)		50.6 ± 10.8	47.9 ± 12.2	0.087
BMI		20 ± 4.1	19.3 ± 4.2	0.20
Hemoglobin (g/dL)		12.1 ± 1.7	11.9 ± 1.8	0.415
RBC counts (10⁶/µL)		4.7 ± 0.8	4.8 ± 0.9	0.215
MCV(fL)		77.2 ± 11.6	74.5 ± 9.5	0.028
MCH (pg)		26.3 ± 4.3	29.7 ± 3.5	0.174
MCHC (g/dL)		33.9 ± 3	33.6 ± 1.9	0.412
Platelet counts (10³/µL)		385.7 ± 148	424.5 ± 156	0.063
White cell counts (10³/µL)		9.9 ± 3.7	10.9 ± 4.3	0.047
Hematocrit (%)		35.6 ± 5.2	35.3 ± 5.3	0.651
Serum ferritin (ng/mL)		402.3 ± 377.2	430.3 ± 397.7	0.60
C-reactive protein		43.1 ± 56.4	44.6 ± 65.6	0.858
Serum vitamin B12		562.0 ± 541.9	570.4 ± 584.1	0.917
Serum folate		8.5 ± 6.2	8.5 ± 6.5	0.988
Total iron-binding capacity		251.6 ± 70.5	241.7 ± 60.7	0.30
Serum transferrin		135.0 ± 45.1	133.3 ± 44.3	0.788
Serum iron		47.7 ± 22.4	46.7 ± 21.8	0.751
Characteristic		Frequency (%)	Frequency (%)	*p*-Value
Sex	Male	105 (79.5%)	27 (20.5%)	0.486
	Female	125 (75.8%)	40 (24.2%)	
History of smoking	Yes	200 (77.5%)	58 (22.5%)	0.269
	No	13 (65%)	7 (3.5%)	
	Ex-smoker	9 (90%)	1 (10%)	
Comorbidities	Yes	189 (78.4%)	52 (21.6%)	0.449
	No	31 (70.5%)	13 (29.5%)	
Country of Birth	Pakistan	221 (77%)	66 (23%)	0.466
	Afghanistan	09 (90%)	01 (10%)	
First language	Chitrali	1 (100%)	0 (0%)	0.745
	Hindko	10 (71.4%)	4 (28.6%)	
	Pashtu	219 (77.7%)	63 (22.3%)	
Religion	Hinduism	1 (100%)	0 (0%)	0.589
	Islam	229 (77.4%)	67 (22.6%)	
Homeownership status	Madrassah	1 (100%)	0 (0%)	0.840
	Own home	178 (77.1%)	53 (22.9%)	
	Rented	51 (78.5%)	14 (21.5%)	
Separate bathroom	Yes	218 (76.8%)	66 (23.2%)	0.190
	No	12 (92.3%)	1 (7.7%)	
Separate kitchen	Yes	191 (94.1%)	12 (5.9%)	0.855
	No	39 (41.5%)	55 (58.5%)	
Type of fuel for cooking	Electricity	2 (100%)	0 (0%)	0.397
	Firewood	65 (78.3%)	18 (21.7%)	
	Natural gas	139 (75.1%)	46 (24.9%)	
	Combined	18 (90%)	02 (10%)	
Sputum AFB	Scanty	78 (98.7%)	1 (1.3%)	0.00
	1+	120 (95.2%)	6 (4.8%)	
	2+	25 (55.6%)	20 (44.4%)	
	3+	7 (14.9%)	40 (85.1%)	

**Table 3 antibiotics-11-01307-t003:** Multivariate regression analysis for association of demographic, anthropometric, and laboratory variables with early or late response (*n* = 288).

Parameter	Univariable Analysis		Multivariable Analysis	
	OR (95% CI)	*p* Value	OR (95% CI)	*p* Value
Age (years)	0.9 (0.9–1.01)	0.215	0.9 (0.9–1.0)	0.473
Sex				
Female	REF		REF	
Male	1.2 (0.7–2.2)	0.438	1.2 (0.4–3.2)	0.222
Smoking Hx				
Non-smoker	REF		REF	
Yes (including ex-smoker)	0.8 (0.3–1.8)	0.571	0.7 (0.2–2.8)	0.505
Co-morbidities				
No	REF		REF	
Yes	0.6 (0.3–1.3)	0.232	0.9 (0.2–3.7)	0.906
BMI				
Low	REF		REF	
Normal	1.3 (0.7–2.4)	0.362	1.7 (0.6–4.7)	0.544
Overweight/obese	0.8 (0.4–1.9)	0.663	0.6 (0.2–2.2)	0.696
Hemoglobin	1.7 (0.9–1.3)	0.413	1.0 (0.7–1.3)	0.520
Platelet count	0.9 (0.9–1.0)	0.067	1.0 (0.9–1.0)	0.614
WBC count	0.9 (0.8–0.9)	0.050	1.0 (0.8–1.1)	0.544
Ferritin	1.0 (0.9–1.0)	0.598	1.0 (0.9–1.0)	0.063
CRP	1.0 (0.9–1.0)	0.855	1.0 (0.9–1.0)	0.338
Vitamin B12	1.0 (0.9–1.0)	0.687	1.0 (0.9–1.0)	0.626
Folate	1.0 (0.9–1.0)	0.131	1.0 (0.9–1.0)	0.172
TIBC	1.0 (0.9–1.1)	0.299	1.0 (0.9–1.0)	0.455
Transferrin	1.0 (0.9–1.0)	0.788	1.0 (0.9–1.0)	0.748
Serum iron	1.0 (0.9–1.0)	0.750	1.0 (0.9–1.0)	0.506
Sputum AFB at baseline				
Scant	REF		REF	
1+	0.3 (0.3–2.2)	0.212	0.3 (0.03–2.4)	0.188
2+	0.02 (0.002–0.12)	<0.001	0.01 (0.001–0.1)	<0.001
3+	0.002 (0.0002–0.02)	<0.001	0.001 (0.0002–0.02)	<0.001

## Data Availability

Data will be made available upon request.

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
