# Peer review of "Determinants of Response at 2 Months of Treatment in a Cohort of Pakistani Patients with Pulmonary Tuberculosis"

_antibiotics, 2022, doi:10.3390/antibiotics11101307_

Round 1
Reviewer 1 Report
Determinants of response at 2-months of treatment in a cohort of Pakistani patients with pulmonary tuberculosis
Dear author and editor,
The article talked about factors associated with early clearance of pulmonary tuberculosis and demonstrated that the early clearance is associated with at – present AFB levels. The article could be published after a minor revision.
I have some comments:
· Could the level of the acid-fast bacilli be indicator for treatment response?
· Did you measure interferon gamma level, What do you think about the measurement of interferon gamma level (Quatiferon test)? could be considered as a better indicator associated with pulmonary TB clearance?
Thank you very much, best regards
Author Response
Dear Reviewer,
Thank you very much for your valuable comments on our manuscript. Following is a point-by-point response to your comments:
- Could the level of the acid-fast bacilli be an indicator for treatment response?
- The level of Acid-Fast Bacilli in sputum smears is unlikely to be a clinically useful indicator of treatment response. We have added these comments to our conclusions.
- Did you measure interferon-gamma level, What do you think about the measurement of interferon-gamma level (Quatiferon test)? could be considered as a better indicator associated with pulmonary TB clearance?
- We did not test Interferon Gamma levels. We have added this point about potential biomarkers of treatment response in the discussion section.
Reviewer 2 Report
The results of study were not quite fresh but it is still a valuable update and reference for readers. I suggest authors answer or do some needful revision to make it more clear.
1.The title of tables should be completed and independently readable. For example, including numbers and participants.
2. It’s more perceived to provide IQR as (Q1,Q3) in Table 1.
3. Table 1: %age seems not correct or wrong typing.
4. Did authors put both age and age categories as independents in multivariable analysis? It was not necessary. Please remove one of them,
5.Line 135-136: The interpretation was not correct. Relapse are patients who have previously been treated for TB, were declared cured or treatment completed at the end of their most recent course of treatment, and are now diagnosed with a recurrent episode of TB. The current study could not determine relapse status because it hasn’t observed patients after cure, but not because all patients were cured. This also means, the following discussion for current study on relapse (138-145) is not suitable and needs to be reorganized.
6. As we know, patients’ medication is also an important factor for treatment outcomes. Authors didn’t mention the medication supervision and taken of patients involved. Please clarify the potential impacts.
Author Response
Dear Reviewer,
Thank you very much for your in-depth review and invaluable comments. Following is a point-by-point response to your comments:
- The title of tables should be completed and independently readable. For example, including numbers and participants.
- Titles of tables have been revised and pertinent details have been added.
- It’s more perceived to provide IQR as (Q1, Q3) in Table 1.
- Tables have been revised to include Q1, Q3 instead of IQR
- Table 1: %age seems not correct or wrong typing.
- Table 1 %ages have been corrected
- Did the authors put both age and age categories as independents in multivariable analysis? It was not necessary. Please remove one of them,
- 'Age categories have been removed from the regression analysis. The table has been updated according to the new analysis. Interpretation has been revised in-text in the results and discussion section.
- Line 135-136: The interpretation was not correct. Relapse are patients who have previously been treated for TB, were declared cured or treatment completed at the end of their most recent course of treatment, and are now diagnosed with a recurrent episode of TB. The current study could not determine relapse status because it hasn’t observed patients after cure, but not because all patients were cured. This also means, the following discussion for current study on relapse (138-145) is not suitable and needs to be reorganized.
- Thank you very much for raising this important point. We have made corrections in the methodology, results, and discussion section according to the correct 'outcome' criteria.
- As we know, patients’ medication is also an essential factor for treatment outcomes. The authors didn’t mention the medication supervision and taking of patients involved. Please clarify the potential impacts.
- DOT therapy was used where a healthcare worker, a school teacher, or an educated neighbor was made an 'observer' for therapy. It was the responsibility of the observer to watch the patient swallow the medicine every time. This detail has been added in the methodology section of the paper.
Round 2
Reviewer 2 Report
Thank you very much for addressing my comments.